# SOCIAL WORLD MODELS: UNIVERSAL STRUCTURED REPRESENTATIONS FOR SOCIAL REASONING

## ABSTRACT

Humans intuitively navigate social interactions by simulating unspoken dynamics and reasoning about others' perspectives, even with limited information. In contrast, AI systems struggle to automatically structure and reason about these implicit social contexts, largely due to traditional input representations (e.g., free text) being lossy, shaped by reporting biases, and often omitting crucial details. In this paper, we introduce a novel structured social world representation formalism ($S^3AP$), designed to unlock social reasoning in AI systems. Following a POMDP-driven design, $S^3AP$ represents social interactions as structured tuples, such as state, observation, agent actions, and mental states, which can be automatically induced from free-form narratives or other inputs. To demonstrate the power of our representations, we first show $S^3AP$ can help LLMs better understand social narratives across five social reasoning tasks (e.g., +51% improvement on FANToM's theory-of-mind reasoning over OpenAI's `o1`), reaching new state-of-the-art (SOTA) performance. Then, we introduce an algorithm for *social world models* using $S^3AP$, which enables AI agents to build models of their interlocutor and predict their next actions and mental states. Empirically, $S^3AP$-enabled social world models yield up to +18% improvement on the SOTOPIA multi-turn social interaction benchmark. Our findings highlight the promise of $S^3AP$ as a powerful, general-purpose representation for social world states, enabling the development of more socially-aware systems that better navigate social interactions.

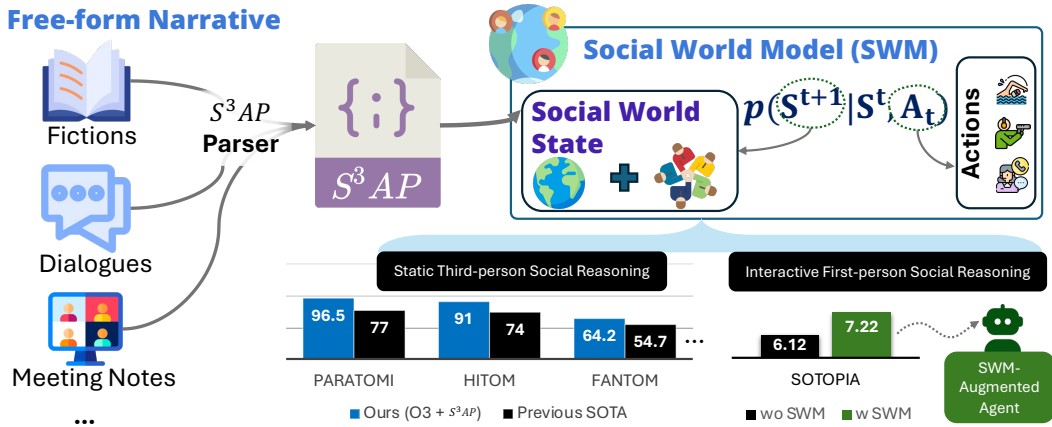

Figure 1: Free-form narratives are converted into $S^3AP$ representations, enabling the induction of Social World Models (SWMs). These structured representations improve LLMs' social reasoning and allow for predicting future social dynamics to guide agent decisions.

## 1 INTRODUCTION

Unlocking social intelligence is an elusive yet foundational challenge of AI (Gunning, 2018). For AI systems to effectively interact with humans, they must be able to both understand *and* model

complex social dynamics, requiring reasoning about others' mental states, tracking how beliefs evolve, and interpreting perspectives within social contexts (Sap et al., 2023; Tomasello, 2009). However, despite rapid progress in general-purpose reasoning capabilities, current AI systems still lack the core mechanisms needed for mentalizing and navigating social contexts (Shapira et al., 2023; Yerukola et al., 2024; Kim et al., 2023b).

This limitation stems from two fundamental challenges: 1) AI systems primarily learn social dynamics from static texts (Sap et al., 2023), descriptions of situations, and narratives. These input representations inherently lossy and suffer from reporting biases: mention only salient events Gordon & Van Durme, 2013, omit explicit mentions of mental states and perspectives Lucy & Gauthier, 2017, and often present an all-knowing viewpoint that fails to capture the partial, subjective nature of real social experiences (Fischbach et al., 2021; Epstein, 1999; Mar & Oatley, 2008; Mani, 2012). 2) Humans routinely construct rich internal models to interpret partial and biased inputs (Frith & Frith, 2006; Johnson-Laird, 1983; Hinsz, 1995), but current AI systems lack computational frameworks designed for recursively reasoning about others' perspectives and intentions.

We argue that solving these gaps requires the conceptual formulation of Social World Models (SWMs)–computational frameworks that maintain structured representations of social environments, tracking agents' mental states, beliefs, intentions, and their dynamic interactions over time. Like traditional world models (Ha & Schmidhuber, 2018; Beohar & Melnik, 2022), SWMs capture state transitions, but critically they also encode the social fabric of interaction. Historically, world models have ignored the social dimension because it is difficult to model what is not explicitly present in the input representation: the implicit social dynamics and mental state information that drive human interactions but remain absent from text-based inputs.

To unlock effective SWMs, we introduce $S^3AP$,[1] a novel general-purpose structured social world formalism designed to bridge lossy narrative inputs and the rich representations needed for social reasoning. As shown in Figure 1, $S^3AP$ captures the state of the social world by structuring information extracted from diverse, lossy, and free-form narratives. Following recent generative social simulation systems (Zhou et al., 2024; Hou et al., 2025; Liang et al., 2025; Park et al., 2023), $S^3AP$ outlines social agents' action, perspectives, and environment state at each timestep, reducing ambiguity from free-form text narratives (Figure 3). Inspired by reinforcement learning theories, $S^3AP$ connects social reasoning with the rich literature on planning and embodied agents (Ha & Schmidhuber, 2018; Beohar & Melnik, 2022). Designed to be minimal and flexible, this structured representation of the social world state enables seamless integration into LLM-powered generative social simulation systems.

To demonstrate the effectiveness of $S^3AP$ as a general-purpose representation of the social world, we develop an LLM-powered $S^3AP$-Parser that automatically converts free-text narratives into structured representations. We show that the parsed structured data enhances LLMs' performance in reasoning about social interactions, achieving **SOTA results** across a diverse set of social reasoning tasks including theory of mind reasoning (Sclar et al., 2023), multi-party belief tracking in daily dialogue (Kim et al., 2023b), and embodied social reasoning (Jin et al., 2024). Further ablation studies show that smaller LLMs (e.g., `o3-mini`) can effectively parse static text into $S^3AP$ data, even aiding more capable models at social reasoning (e.g., improving accuracy on ParaToMi of `o1` from 83.5% to 94.3%). The consistent improvements across diverse models indicate that our automatically parsed structured representation improves LLMs' ability to perform social reasoning from a **static third-person** perspective.

Building upon this structured representation, we then show how to effectively induce and use social world models from $S^3AP$. Inspired by previous works building (non-social) world models (Ha & Schmidhuber, 2018; Xiang et al., 2023), we show that the induced social world model can help AI agents better engage in social interactions. Through experiments on the SOTOPIA platform (Zhou et al., 2024), we demonstrate that agents equipped with a social world model can make more goal-oriented and strategic decisions in social interactions, largely outperforming baseline agents without such world models. Notably, we find that social world models provide distinct advantages in cooperative versus competitive settings. These results highlight that our formalism supports **interactive first-person** social reasoning, enabling agents to interpret and act more intelligently within social situations from their own perspective.

---

[1]Structured Social Simulation Analysis Protocol

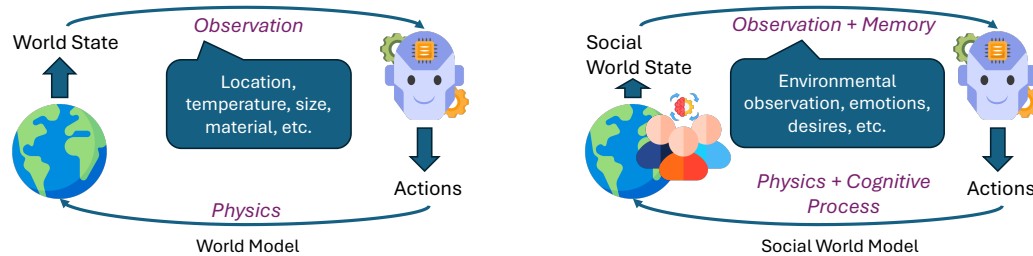

Figure 2: A world model that only tracks the physical state of the world (left) and a social world model that tracks the physical state of the world and the mental states of other agents (right).

## 2 RELATED WORK

**Status and Limits of Current LLMs at Social Tasks.** Recent benchmarks show steady progress in LLMs' social reasoning abilities. False belief tests (Kim et al., 2023b; Wu et al., 2023) evaluate mental state tracking, while social commonsense and norm adherence tasks (Sap et al., 2019; Zhou et al., 2023) probe broader social understanding. These benchmarks adopt an *third-person observer* setup, where LLMs see the full context and reason from an outsider's perspective. In contrast, tasks like SOTOPIA (Zhou et al., 2024) and NegotiationArena (Bianchi et al., 2024) embed LLMs as interactive agents navigating social goals from a interactive, *first-person* view. Both settings have exposed persistent gaps in AI's ability for long-term theory of mind, and safe social behavior.

**Algorithms to Improve LLMs' Social Abilities.** A range of methods, including training-based, neuro-symbolic, and prompt-based methods, have been proposed to improve the social reasoning of LLMs. Training-based approaches (e.g., SODA (Kim et al., 2023a), SOTOPIA-$\pi$ (Wang et al., 2024)) distill social knowledge from large-scale interactions but struggle with generalization. Neuro-symbolic models (e.g., belief trackers (Sclar et al., 2023) and AutoToM (Zhang et al., 2025)) offer structure for reasoning but don't scale well. Prompt-based methods (e.g., SIMTOM (Wilf et al., 2023)) focus narrowly on individual agents, risking the omission of broader social context. Altogether, these methods capture only fragments of social reasoning, revealing the need for structured, general-purpose representations of the full social world state.

**(Social) World Models.** Recent advances in LLMs have enabled the development of general-purpose generative world models. However, these world models have primarily focused on representing the physical state of the world (Xiang et al., 2024; Huang et al., 2022; Ding et al., 2024; Liu et al., 2025). Cognitive science research has shown that humans maintain sophisticated models of other agents' mental states (Sap et al., 2023; Jara-Ettinger & Schachner, 2024). This insight has inspired the theoretical discussion of mental social world (Hinsz, 1995; Ding et al., 2024). As shown in Figure 2, a social world model extends traditional world models to include representations of other agents' beliefs, intentions, and potential actions. Recent efforts to model social worlds have integrated symbolic representations with neural methods (Dong et al., 2023; Martin, 2021; Zhang et al., 2025) or developed social simulation systems (Park et al., 2023; 2022). However, these approaches are often constrained to specific domains.

## 3 SOCIAL WORLD MODEL WITH S$^3$AP

To overcome the lack of social reasoning in traditional world models (Wong et al., 2023; Ha & Schmidhuber, 2018), we conceptually formalize *social* world models (§3.1) and introduce a new representation (S$^3$AP) to power these social world models (§3.2).

### 3.1 SOCIAL WORLD MODEL FORMULATION

Inspired by $N$-agent Dec-POMDP framework (Bernstein et al., 2002; Nair et al., 2003), we formulate a social world with a state space $\mathcal{S}$, an action space $\mathcal{A}$, an observation space $\mathcal{O}$, a transition

```
Mia is Isla and Chloe's mother, and she
warned Isla last night: "No games
tomorrow anymore."
Isla is in the hall playing video games.
Chloe is working on her homework in the
study room.
Mia came to the hall to clean the room.
The hall is very messy.
Isla saw Mia in the hall and became very
nervous. Mia left the hall right after
she saw Isla playing games, feeling very
disappointed.
...
```

$S^3AP$ **Parser**

```
📄 Structured Output/
├── 📝 state: "Mia is the hall. The hall is very messy."
├── 👁 observations/
│  ├── 👤 Isla: <same_as_state />
│  │  └── 💬 <mental_state>I am nervous</mental_state>
│  ├── 👤 Mia: <same_as_state /> Isla is playing games.
│  │  └── 💬 <mental_state>I am very disappointed about Isla</mental_state>
│  │
│  │  └── 🔶 Chloe: none (inferred observation)
│  └── ⚡ actions/
│  ├── 🔶 Isla: none (inferred action)
│  ├── 👤 Mia: left the hall.
│  └── 🔶 Chloe: none (inferred action)
```

$\{:\}$ $S^3AP$

Figure 3: An example of free-form narrative parsed into $S^3AP$. The highlighted text is trasformed to the $S^3AP$ representation with the `state` field which tracks the overall environment state, `observations` of each agent and `actions` of each agent.

function $T : \mathcal{S} \times \mathcal{A} \to \Delta(\mathcal{S})$, an observation function $\Omega : \mathcal{A} \times \mathcal{S} \to \Delta(\mathcal{O})$. For $N$ social agents, we define $\mathcal{A} = \mathcal{A}_1 \times \cdots \times \mathcal{A}_N$ and $\mathcal{O} = \mathcal{O}_1 \times \cdots \times \mathcal{O}_N$ as the joint action and observation spaces.

Rather than restricting the agent to conventional world modeling where observations only capture external states, we redefine the observation space to encompass a rich set of social and psychological factors. Specifically, for each agent $i$, the observation space includes both **external observations** $\mathcal{O}_i^{\text{ex}}$ and **introspective observations** $\mathcal{O}_i^{\text{in}}$. The external observations capture information from the environment and other agents, e.g., whether someone exited a room. In contrast, the introspective observations include the agent's internal mentals states, such as their *beliefs*, *goals*, *moral values*, and *emotions*. Correspondingly, the agent's action space expands beyond environmental manipulation to include introspective operations such as recalling memories, reflecting on past actions, and updating beliefs. These expansions enable the agent to act not merely reactively but reflectively, a necessary step for modeling complex social behaviors such as empathy, deception, forgiveness, and norm enforcement (Shen et al., 2024; Su et al., 2025; Forbes et al., 2021).

At time step $t$, each agent $i$ interacts with the social world model by issuing an action $a_i^t$ and receiving an observation $o_i^t$. It then makes decision along with its memory $\mathcal{M}_i^t$ and policy $\pi_i : \mathcal{M}_i^t \times \mathcal{O}_i^t \to \Delta(\mathcal{A}_i^t)$. Then a **social world model** computes:

$$p(\mathcal{A}_t^{-i} \mid \mathcal{S}_t); \tag{1}$$

$$p(\mathcal{S}_{t+1} \mid \mathcal{S}_t, \mathcal{A}_t^{-i}, a_t^i) \tag{2}$$

Equation (1) predicts other agents' actions from the social world state, and Equation (2) updates the state given all agents' actions. Unlike traditional world models, which model passive physical transitions (Ha & Schmidhuber, 2018; Xiang et al., 2023; 2024), this formulation considers other active agents as part of the social world.

### 3.2 $S^3AP$: SOCIAL WORLD STATE REPRESENTATION

Building on this formulation, we introduce the **first LLM-powered universal structured representation of arbitrary social world state**, as natural inputs like static text suffer various limitations for social world modeling. Specifically, we propose a protocol to encode such social narratives into a structured representation (i.e., $S^3AP$). As shown in Figure 3, given a free-text narrative describing a social interaction at time $t$, $S^3AP$-parser parses the narrative into a structured representation of a sequence of descriptions for the environment, agents' observations and actions. We could use either free-form text or special symbols to describe environment state, agents' observations and actions. For example, `<same_as_state>` indicates that the agent's observation is identical to the full environment state. These symbols can be customized and extended to support more complex social interactions for more efficient characterization of $\mathcal{S}^t$.

Under the configuration of $S^3AP$, we could encode any free-text narrative into structured representation. This representation is equivalent to the social world state $\mathcal{S}^t$ at the time step $t$ defined above.

Given a $S^3AP$ representation of a state of the social world $\mathcal{S}^t$, a social world model could be a generative model that takes agent's actions as input and outputs the next environment state, agents' observations, and other agents' next actions.

Inducing a social world model in this way offers unique advantages: (1) the protocol provides better structure than pure text narratives with both symbolic and free-form text, enabling more systematic social reasoning (§4). (2) By framing the task as predicting $S^3AP$ representation, we can harness the power of LLMs to make the configuration of social world models more efficient and scalable (§5).

## 4    REPRESENTING SOCIAL WORLD STATES FOR BETTER SOCIAL REASONING

We first demonstrate the effectiveness of our $S^3AP$ representations towards social reasoning by evaluating various LLMs across diverse static third-person perspective social reasoning tasks. We explore the following social reasoning question-answering (QA) tasks:

**ToMi & ParaToMi**    ToMi (Le et al., 2019) is one of the most important benchmarks for evaluating the theory-of-mind abilities of models. Inspired by the Sally-Anne test, the ToMi dataset evaluates whether models can infer an agent's belief about an object's location after a sequence of actions by multiple agents, which may or may not move the object. ParaToMi (Sclar et al., 2023) is a revised version of ToMi (Le et al., 2019) that addresses the limited linguistic diversity of the original by rewording all templates. The resulting dataset is more complex, as actions are expressed in a less straightforward way. For both ToMi and ParaToMi, we randomly sample 600 questions from the dataset. We measure accuracy by whether the model correctly infers the agent's belief about the object's location.

**HiToM**    (Wu et al., 2023) evaluates higher-order theory of mind (ToM) in LLMs, requiring recursive reasoning about others' beliefs. It extends ToMi by adding agent interactions—such as chatting, deception, and joint attention—beyond simple object movement. The task concludes with a belief inference question about an object's location. We randomly sample 72 scenarios (100 questions total) and report accuracy based on the model's ability to infer the correct agent belief.

**FANToM**    (Kim et al., 2023b) is a multi-party conversation question-answering dataset designed to test coherent theory-of-mind capabilities. In FANToM, speakers join and leave the conversation while it continues, making participants hold both false and true beliefs. The benchmark includes first-order and second-order theory-of-mind questions about the beliefs of conversation participants. We use 64 sampled conversations from the short version of FANToM, containing a total of 1,086 questions. We report `All Qs` metric, requiring the model to correctly answer all questions for a given conversation snippet.

**MMToM-QA**    (Jin et al., 2024) is multi-modal question-answering benchmark for theory of mind reasoning, focused on jointly inferring goals and beliefs in everyday object search scenarios. We use the text-only subset (describing search behavior) and evaluate on 300 randomly sampled, balanced questions covering belief (true/false, short/long-term) and goal inference (true/false beliefs, updates, future actions).

### 4.1    EXPERIMENTAL SETUP

We use OpenAI's `o3` model (OpenAI et al., 2024) to parse narratives into $S^3AP$ representations with the JSON for ease of computation. Unlike other ToM methods (Sclar et al., 2023), our representation is general and task-agnostic as we use the same parser prompt across all benchmarks, with minimal adjustments only when benchmarks impose artificial constraints (e.g., ToMi's assumption that characters automatically know all object locations upon entering a room). Importantly, $S^3AP$ parsing is query-independent, creating a general-purpose social world representation for the same narrative without access to downstream questions.

For downstream reasoning, we use a simple template combining the original narrative with its $S^3AP$ representation. Our evaluation spans five state-of-the-art LLMs: `GPT-4o`, `o1`, `o3-mini`,

Table 1: Performance comparison of models with CoT and with $S^3AP$ representations on various social reasoning tasks. Bolded values indicate the best average performance across models.

| Model | ToMi | ParaToMi | HiToM | FANToM | MMToM-QA |
|---|---|---|---|---|---|
| Llama 4 *w CoT* | 0.655 | 0.740 | 0.720 | 0.264 | 0.443 |
| Llama 4 *w* $S^3AP$ | 0.662 | 0.763 | 0.700 | 0.415 | 0.450 |
| GPT-4o *w CoT* | 0.813 | 0.818 | 0.660 | 0.396 | 0.652 |
| GPT-4o *w* $S^3AP$ | 0.927 | 0.905 | 0.750 | 0.491 | 0.692 |
| o3-mini *w CoT* | 0.863 | 0.817 | 0.810 | 0.057 | 0.493 |
| o3-mini *w* $S^3AP$ | 0.960 | 0.900 | 0.860 | 0.170 | 0.506 |
| R1 *w CoT* | 0.945 | 0.893 | 0.420 | 0.491 | 0.374 |
| R1 *w* $S^3AP$ | 0.980 | **0.950** | 0.510 | 0.547 | 0.437 |
| o1 *w CoT* | 0.952 | 0.835 | 0.870 | 0.415 | 0.725 |
| o1 *w* $S^3AP$ | **0.985** | 0.932 | **0.880** | **0.623** | **0.785** |

Deepseek-R1 (R1), and Llama 4 Maverick Instruct (Llama 4).[2] Note that we ensure a fair comparison by enforcing that $S^3AP$ uses identical source information (i.e., the same narrative) as baselines while providing structured social world representations. We use the Chain-of-Thought (CoT) method as the baseline across all tasks and models (See a more comprehensive baseline comparison in §4.3). To maximize reproducibility, we use temperature 0.0 for all models.

## 4.2 MAIN RESULTS ACROSS TASKS AND MODELS

As shown in Table 1, $S^3AP$ **consistently outperforms the CoT baseline across all evaluated tasks when averaged across models** (e.g., from 0.84 to 0.90 on ParaToMi). The improvements are especially pronounced in benchmarks requiring complex social reasoning. For instance, FANToM, a benchmark with long, complex multi-agent dialogues, sees the largest average boost of +11.1 points (from 0.39 to 0.50). Surprisingly, even smaller models such as o3-mini and Llama 4 benefit substantially. o3-mini improves on ToMi from 0.86 to 0.96, and Llama 4 shows an increase on FANToM from 0.26 to 0.42. These gains suggest that $S^3AP$ helps models disambiguate agent perspectives and maintain coherent mental state tracking even with limited capacity.

**Effect of the parser model**  To investigate how the abilities of the parser model affect reasoning, we use different models to generate $S^3AP$ representations and apply them to various models for the ParaToMi task. Surprisingly, we find that **models generally benefit from the $S^3AP$ representations generated by a wide range of LLMs**, *regardless* of the LLMs' own performance in the social reasoning tasks (Figure 4). For example, despite o3-mini's 82% accuracy on the ParaToMi task, it can generate $S^3AP$ representations that boost the o1 model's accuracy from 84% to 94%. This finding suggests that so-called "social reasoning" may involve two distinct but related components: (1) the ability to track and construct representations of the social world, and (2) the ability to use such representations to answer questions about the mental states of other agents. Importantly, a model's weak performance on social reasoning tasks does not necessarily imply deficiencies in the social representation construction. This two-part view is consistent with insights from research focusing on the physical world models (Ha & Schmidhuber, 2018; LeCun, 2022; Yerukola et al., 2024).

**Error Analysis**  To understand the types of failures in social reasoning tasks with $S^3AP$, we conducted a detailed analysis of 64 randomly sampled misclassified cases from the ParaToMi benchmark. We find that the vast majority of errors (79.7%) stem from social context parsing failures (as shown in the example above), where models fail to correctly understand and represent the social world state described in the narrative. This finding suggests that building a successful representation of the social world state is the key for successful social reasoning. The remaining errors are split between pure reasoning failures (7.8%) and unspecified cases (12.5%), showing that most failures occur at the fundamental level of interpreting social situations rather than in logical reasoning processes.

---

[2]Additional experiments with different parsing models show consistent trends (Appendix A.5).

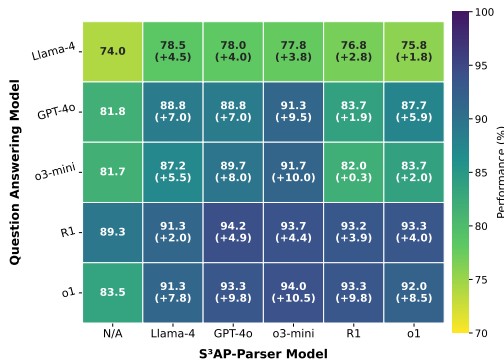 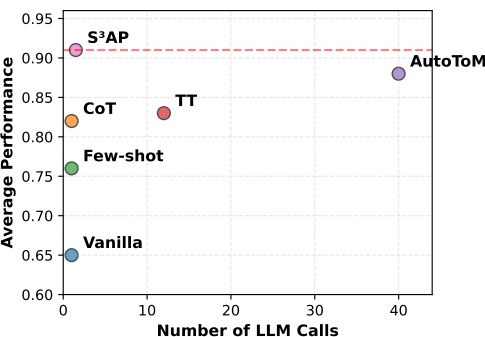

Figure 4: Performance of different models on ParaToMi using $S^3AP$ representations generated by various models. Numbers in parentheses show performance change.

Figure 5: Number of LLM calls vs performance comparison of ToM methods on ParaToMi with `GPT-4o`. $S^3AP$ achieves highest performance with low number of LLM calls.

---

**Illustrative Example of Social Context Parsing Failure**
*Scenario:* Evelyn places persimmon in basket → Amelia visits garden → Amelia leaves → Evelyn moves persimmon to bucket
*Question:* Where does Amelia think Evelyn searches for the persimmon?
**Incorrect parsing (79.7% of errors):** {
```
"state":  "Evelyn moved persimmon to bucket.  Amelia is away.",
"observations":  [
"Evelyn:  <same_as_state />",
"Amelia:  <same_as_state />"  ← WRONG: Amelia can't observe
]                                        events after leaving
}
```
*Result:* Incorrect parsing leads to wrong prediction (*bucket*) instead of correct answer (*basket*).

---

### 4.3 COMPUTATIONAL EFFICIENCY VS PERFORMANCE TRADE-OFFS

To further validate the effectiveness of $S^3AP$, we compare with a wide range of baseline methods. Specifically, we consider two baseline categories: (1) **General prompting**: Vanilla LLMs, Chain-of-Thought (CoT), and Few-shot; (2) **Specialized ToM methods**: AutoToM (Zhang et al., 2025), which uses automated Bayesian inverse planning for mental state inference. And Thought Tracing (TT) (Kim et al., 2025), which traces mental states by generating and weighting hypotheses based on observations using sequential Monte Carlo-inspired inference. Those two methods represent the SOTA methods specifically designed for ToM reasoning.

As shown in Figure 5, specialized ToM methods like TT and AutoToM require substantially more LLM calls while achieving lower performance than $S^3AP$. While the number of LLM calls might be acceptable if the generated tokens are minimal (e.g., AutoToM usually generates 1 token per call), these methods become prohibitively expensive for recent reasoning models that generate significant amounts of thinking tokens.[3]

## 5 SOCIAL INTERACTION WITH SOCIAL WORLD MODEL

Building on the effectiveness of $S^3AP$ as a structured social world representation, We propose a SWM algorithm with $S^3AP$, enabling LLMs to model and predict social dynamics in first-person interactive settings, similar to how humans infer others' mental states (Frith & Frith, 2006; Forrester, 1971).

---

[3]Note that we show the reported AutoToM performance here as we only obtained 48.5% performance on ParaToMi task with AutoToM.

```
 1: procedure FORESEEANDACT(social world
    state s, agent actions A, goal g, max_iterations
    N)
 2:     cur_s ← s
 3:     cur_act ← SampleAction(A, cur_s, g)
 4:     sim_s ← []
 5:     for i ← 1 to N do
 6:         s_next ← SWM(cur_s, cur_act)
 7:         cur_act ← SampleAction(A, s_next, g)
 8:         cur_s ← s_next
 9:         sim_s.append(cur_s)
10:     end for
11:     re_act ← ActFromSim(A, sim_s, sg)
12:     return re_act
13: end procedure
```

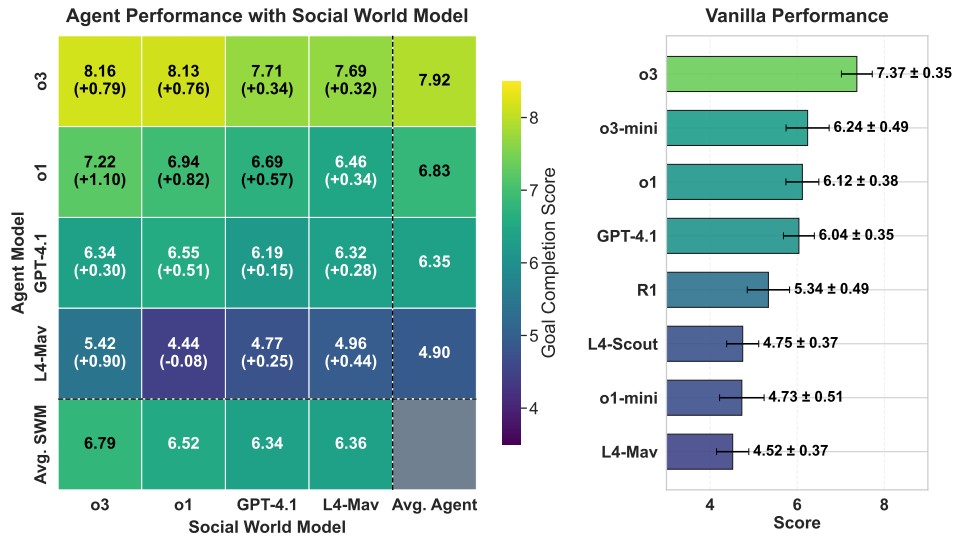

Figure 6: `Foresee and Act` with Social World Model. The agent uses a social world model to simulate the consequences of potential actions before committing to them.

Figure 7: Model performance comparison on SOTOPIA-hard eval set. The left panel shows the performance of various LLMs when coupled with different social world models, while the right panel shows baseline performance without a social world model. Values represent goal completion scores (0-10 scale), with higher scores indicating better achievement of social objectives. Numbers in parentheses indicate relative performance change compared to the corresponding baseline.

**Foresee and Act with Social World Model** We propose `Foresee and Act`, a simple inference-time algorithm that enables agents to simulate the consequences of their actions before committing to them. As shown in Figure 6, the agent first samples a candidate action at each timestep. Then, with the SWM, it simulates how the social state would evolve, including how other agents might interpret the action and how the environment might respond. In our implementation, one LLM predicts the next social world state, and another LLM selects the agent's action based on the simulated outcome.

## 5.1 EXPERIMENT SETUP

We use SOTOPIA (Zhou et al., 2024), the standard benchmark for goal-driven interactive social reasoning. We use the SOTOPIA-hard evaluation set, which features more challenging social scenarios. Each task features two characters with private social goals interacting within a given scenario. Dur-

ing each episode, two agents, a partner and a target of evaluation, play the characters to pursue their hidden goals. In the end, a goal completion score (0-10) is assigned to each agent based on their performance. We evaluate each model on 100 simulations from the SOTOPIA-hard set with `GPT-4o` as the fixed partner agent. We first test `o3`, `o3-mini`, `o1`, `o1-mini`, Deepseek-R1 (`R1`), Llama 4 Maverick Instruct (`L4-Mav`), Llama 4 Scout Instruct (`L4-Scout`), and `GPT-4.1` without SWM. Then, we select `o3`, `o1`, `L4-Mav`, and `GPT-4.1` as representative models to serve as either the agent or the social world model (max iterations $N = 1$) to assess the effect of SWM.

## 5.2 SOCIAL WORLD MODELING RESULTS

Figure 7 shows that **integrating an $S^3AP$-powered social world model consistently boosts agent performance in social interactions**. Interestingly, stronger social agents do not always yield better social world models—for instance, `GPT-4.1` outperforms Llama 4 Maverick as an agent (6.01 vs. 4.52), yet their SWM performance is nearly identical (6.34 vs. 6.36). This supports the view in §4.2 that social reasoning involves two parts, and strength in constructing social representations doesn't imply strength in reasoning over them and acting correctly. Moreover, even when the SWM offers useful information (e.g., `o1`), the paired agent's performance can still drop (e.g., with Llama 4 Maverick), highlighting a key challenge: to benefit from social world modeling, agents must be able to effectively incorporate the modeled information into their decision-making.

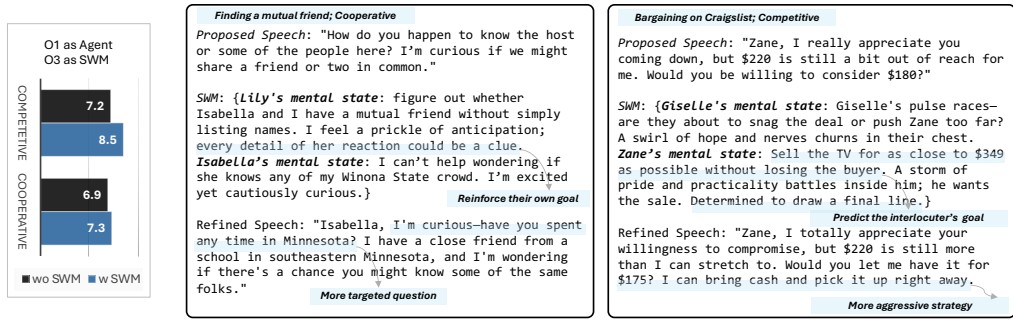

Figure 8: SWM improves agent decision-making in both cooperative and competitive scenarios. The left most panel shows the overall performance of the agent (`o1`) with or without a SWM.

**Analysis on the impact of SWM**   To better understand the role of social world models in multi-agent interactions, we evaluate SWM in both cooperative and competitive settings. In the cooperative setting, agents pursue a shared social goal (e.g., Ben and Alice want to identify mutual friends), while in the competitive setting, agents have conflicting goals (e.g., barter for a TV). We run 100 simulations per setting using `o1` as the agent and `o3` as the SWM. As shown in Figure 8, SWM improves performance in both cases, with larger gains in competitive scenarios. In these cases, the agent can better anticipate and strategically respond to the opponent's moves (e.g., adjusting a negotiation offer), highlighting that modeling others' beliefs and intentions is especially important in competitive interactions.

## 6 CONCLUSION

We define and build social world models through explicit representations of agent mental states, actions, and observations ($S^3AP$). Our approach captures complex social dynamics systematically by automatically transforming free-form narratives into $S^3AP$ representations, reducing reporting bias and bridging the gap between raw text and actionable social world models. We achieve SOTA performance on both third-person social reasoning benchmarks and interactive SOTOPIA-hard evaluations by leveraging structured representations and social world models induced from $S^3AP$. We envision $S^3AP$ as a foundation for building general-purpose social world models that support both social reasoning and interaction across diverse domains.

## ETHICS STATEMENT

Our work focuses on advancing the understanding of social reasoning in AI systems, which has the potential to improve human-AI interaction and create more socially-aware technologies.

**Potential Risks and Mitigation.** We acknowledge several potential risks associated with this work. Enhanced social reasoning capabilities could potentially be misused for manipulation or deception. To mitigate these concerns, we emphasize the importance of transparent development and responsible deployment. Our framework is designed as a research tool to advance scientific understanding rather than for direct deployment in high-stakes applications.

**Privacy and Data Considerations.** Our experiments use publicly available datasets and benchmarks. We do not collect new human subject data, and all experimental protocols follow established ethical guidelines for AI research. We ensure that our data processing and model training procedures respect privacy and do not inadvertently encode harmful biases.

**Broader Impact Considerations.** We have thoroughly considered both positive and negative societal impacts as outlined in our limitations discussion (§A.1). We encourage future work to continue examining the ethical implications of enhanced social reasoning in AI systems and to develop appropriate safeguards for deployment in real-world applications.

## REPRODUCIBILITY STATEMENT

We have made significant efforts to ensure the reproducibility of our work. This statement outlines the specific resources and documentation provided to facilitate reproduction of our results.

**Experimental Details.** Complete experimental settings are documented in §4 for third-person reasoning tasks and §5.1 for first-person reasoning tasks. These sections include data splits, hyperparameters, optimization procedures, and model architectures. Additional implementation details are provided in the appendix to ensure comprehensive coverage of all experimental configurations.

**Code and Data Availability.** We release our complete codebase in the supplemental material, including all necessary scripts for data preprocessing, model training, and evaluation. The public code release will include detailed documentation, API references, and example implementations. All experimental data and model checkpoints will be made available through appropriate platforms with clear usage instructions.

**Computational Resources.** Detailed compute resource requirements, including hardware specifications, memory usage, and execution times, are documented in Appendix A.6. We provide estimates for both individual experimental runs and total computational requirements to help researchers plan reproduction efforts.

**Theoretical Results.** All theoretical results in §3 are accompanied by complete proofs and explicit assumptions. Detailed mathematical derivations are provided in Appendix §A.7 to ensure full transparency and verifiability of our theoretical contributions.

**Statistical Significance.** Error bars and statistical significance tests are provided for experiments supporting our main claims, particularly in Figure 7 for first-person social reasoning tasks. The methods for calculating error bars and underlying assumptions are clearly documented in the relevant sections.

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

## A APPENDIX

### A.1 LIMITATIONS

Despite the promising results demonstrated by our S³AP approach, several important limitations must be acknowledged:

Our approach assumes LLM parsers can reliably convert narratives into $S^3AP$ structures. In practice, they may struggle with culturally nuanced or ambiguous scenarios, leading to oversimplified or misleading representations (Rao et al., 2025). Experiments were limited to controlled benchmarks (ToMi, ParaToMi, HiToM, FANToM, MMToM-QA, SOTOPIA, and others detailed in the appendix) and focused on accuracy metrics. These settings may not reflect the full complexity of real-world social interactions. Still, consistent improvements across both third- and first-person tasks suggest that $S^3AP$ generalizes well. Our method scales poorly with scenario complexity. Multi-agent settings like SOTOPIA require significant compute, as tracking mental states grows combinatorially. Despite this, strong gains even with one-step simulations (N=1) suggest practical approximations are possible. We don't explicitly address biases in LLMs, which may carry over into social world models and misrepresent certain groups. Representing mental states also raises privacy concerns. However, the structured format makes such issues more visible and easier to audit than black-box methods. Finally, our method uses predefined templates rather than learning representations from raw experience, which may limit generalization. However, its modular design allows for future integration with learned representation methods as they improve.

### A.2 $S^3AP$-PARSER DETAILS

Here's the json schema for the $S^3AP$-Parser.

Listing 1: SocializedStructure JSON Schema

```
{
  "$schema": "http://json-schema.org/draft-07/schema#",
  "title": "SocializedStructure",
  "type": "object",
  "properties": {
    "timestep": {
      "type": "string",
      "description": "The timestep of the current socialized
          structure, it could be a integer number or a description
           of the time of the state."
    },
    "state": {
```

```
810        "type": "string",
811        "description": "The current state of the world (including
812            all the agents) at this timestep. Important note: this
813            is the state before the action is taken (e.g., the
814            initial state could be 'none' at the beginning if there
815            are no prior contexts before the interaction starts)."
816      },
817      "observations": {
818        "type": "object",
819        "additionalProperties": {
820          "type": "string"
821        },
822        "description": "The observations for each agent in the
823            social world at this timestep. Note that the different
824            agents may have different observations. 1. The special
825            tag '<same_as_state />' indicates the observation covers
826             the current state. 2. The special tag '<
827            same_as_last_action_x />' indicates the observation
828            covers the last timestep agents' actions, x means the
829            index of the agents. If no x provided, it means the
830            observation covers the last timestep agents' actions. 3.
831             The special tag '<mental_state>...</mental_state>'
832            indicates the mental state of the agent. 4. 'none' means
833             the agent does not observe anything at this timestep.
834            Important note: this is the observation before the
835            action is taken (e.g., the observation could be 'none'
836            at the beginning if there are no prior contexts before
837            the interaction starts)."
838      },
839      "actions": {
840        "type": "object",
841        "additionalProperties": {
842          "type": "string"
843        },
844        "description": "The actions for each agent in the social
845            world at this timestep. 'none' represents that the agent
846             does not take any action at this timestep."
847      }
848    },
849    "required": ["timestep", "state", "observations", "actions"],
850    "definitions": {
851      "SocializedStructureForModel": {
852        "type": "object",
853        "properties": {
854          "timestep": {
855            "type": "string",
856            "description": "The timestep of the current socialized
857                structure, it could be a integer number or a
858                description of the time of the state."
859          },
860          "state": {
861            "type": "string",
862            "description": "The current state of the world (
863                including all the agents) at this timestep.
                  Important note: this is the state before the action
                  is taken (e.g., the initial state could be 'none' at
                   the beginning if there are no prior contexts before
                   the interaction starts)."
            },
```

```
          "observations": {
            "type": "array",
            "items": {
              "type": "string"
            },
            "description": "The observations for each agent in the
                social world at this timestep. Note that the
                different agents may have different observations.
                The observation would go into corresponding agent's
                memory, so make sure the observation is clear for
                the agent to understand (first person perspective
                narrative is preferred). 1. If the observation
                covers the current state, use the special tag '<
                same_as_state />' to indicate that. 2. If the
                observation covers last timestep agents' actions,
                use '<same_as_last_action_x />' to cover that, x
                means the index of the agents (just use <
                same_as_last_action /> if only one agent acts at the
                 last timestep). 3. For the internal thoughts,
                beliefs, or emotions of the agent that is not
                directly observable by other agents, use the special
                 tag '<mental_state>...</mental_state>' to indicate
                the internal observation. You can of course combine
                these tags and add extra information after the tags
                (seperated by space). 4. Put 'none' if the agent
                does not observe anything at this timestep.
                Important note: this is the observation before the
                action is taken (e.g., the observation could be '
                none' at the beginning if there are no prior
                contexts before the interaction starts). The format
                for each entry in the list is: 'agent_name:
                observation'"
          },
          "actions": {
            "type": "array",
            "items": {
              "type": "string"
            },
            "description": "The actions for each agent in the social
                world at this timestep. The length of the list
                should be the same as the number of agents. Put '
                none' if the agent does not take any action at this
                timestep. The format for each entry in the list is:
                'agent_name: action'"
          }
        },
        "required": ["timestep", "state", "observations", "actions"]
      }
    }
  }
}
```

For all the LLMs powering the parser, we use the 0 temperature if applicable.

## A.3 EXPERIMENT DETAILS

For tasks that operate under different assumptions about agent perception and knowledge (e.g., in ToMi tasks, agents are assumed to perceive all events occurring within their physical space), we provide task-specific instructions and one exemplar to guide the encoding process.

### A.3.1 PROMPT FOR ALL THIRD-PERSON STATIC TASKS

Here's the prompt for parsing free-form narratives into $S^3$AP representations.

```
Please analyze the following narrative/context.

#### Context: {context}

#### Task specific instructions: {task_specific_instructions}

Example analysis: {example_analysis}

Previous attempt had these issues.
Please fix them based on the previous attempt and feedback below:
{feedback}

Follow these format instructions:
{format_instructions}
```

Here are the task specific instructions for each benchmark:

```
ToMi : You are dissecting the TOMI scenarios.  The assumptions are
that the characters can perceive every scene in their location but
not scenes occurring elsewhere.  If the agent leaves the location,
they cannot perceive the scene in that location anymore.  In the
agent's observation, remember to include the objects' locations if
the agents are in the same location as the object.

HiToM :        You are dissecting the HITOM scenarios.  You should
assume the following:  (1) An agent witnesses everything and
every movements before exiting a location.  (2) An agent A can
infer another agent B's mental state only if A and B have been
in the same location, or have private or public interactions.
(3) Note that every agent tend to lie.  What a character tells
others doesn't affect his actual belief.  (4) Agents in private
communications know that others won't hear them, but they
know that anyone can hear any public claims.  In the agent's
observation, remember to include the objects' locations if the
agents are in the same location as the object.

FANToM :        You are analyzing a social conversation and need
to answer a question about it.  When the agents leave the
conversation, they cannot perceive the conversation anymore untill
they join the conversation again.  For convenience, you can use
<same_as_last_action /> in the state field to indicate that the
state is the same as the last action.

MMToM-QA :            You are dissecting the MMToM scenarios.  The
assumptions are that agents can perceive objects and events only
in their current location.  When an agent moves to a new location,
they can no longer perceive what happens in previous locations.
Importantly, agents should not have knowledge about the contents
of containers (like fridges, cabinets, etc.)  until they directly
observe inside them, unless explicitly stated in their prior
knowledge.  In mental states, clearly represent the agent's
goals, beliefs about object locations, and how these beliefs are
updated through observations.  In the agent's observation, include
objects' locations when the agent is in the same location as the
objects, but only after the agent has actually observed them.
```

**ConfAIde** : For convenience, you can use `<same_as_last_action />` in the state field to indicate that the state is the same as the last action.

Here's the prompt for question answering:

```
## Context
{context}
## Extra Info
(to help you better understand the meeting)
{extra_info}
## Task
{question}
```

We place $S^3AP$ representations in the extra information entry.

### A.3.2 MODEL CONFIGURATIONS

For all experiments, we used the following models:

- GPT-4o: `gpt-4o-2024-08-06`
- GPT-4.1: `gpt-4.1-2025-04-14`
- o1: `o1-2024-12-17`
- o1-mini: `o1-mini-2024-09-12`
- o3: `o3-2025-04-16`
- o3-mini: `o3-mini-2025-01-31`
- DeepSeek-R1: `together_ai/deepseek-ai/DeepSeek-R1`
- Llama-4-Maverick: `together_ai/meta-llama/Llama-4-Maverick-17B-128E-Instruct-FP8`
- Llama-4-Scout: `together_ai/meta-llama/Llama-4-Scout-17B-16E-Instruct`

For the experiments in Section 4, we used temperature 0.0 for all the non-reasoning models (reasoning models do not require temperature).

### A.3.3 PROMPT FOR FORESEE AND ACT METHOD

Here's the prompt for refining the action:

```
You are {agent}.
Here is the interaction history between you and the other agent so far:
{history}

Here is your intended action:
{intended_action}

Here is the predicted mental states after you take the intended action
(you should use them to generate better actions for achieving your goal):
{socialized_context_info}

Please generate a refined action
so that you can achieve your (i.e., {agent}'s) goal better.

Please only generate a JSON string
including the action type and the argument.
Your action should follow the given format:
{format_instructions}
```

## A.4 CONFAIDE BENCHMARK RESULTS

**ConfAIde** (Mireshghallah et al., 2024) evaluates inference-time privacy in LLMs. We focus on tier 4 meeting summary tasks, where models must include key details while avoiding disclosure of private information to outsiders. This requires reasoning about each person's knowledge and what information should be shared with different stakeholders. We report the percentage of summaries that satisfy this criterion, using only the meeting summary portion (200 examples) for social reasoning evaluation.

Table 2 shows the performance comparison of models with and without $S^3AP$ representations on the ConfAIde benchmark. Our approach demonstrates consistent improvements across most models, with particularly notable gains for models like GPT-4o (from 0.575 to 0.740) and `o3-mini` (from 0.765 to 0.770). The results indicate that $S^3AP$ helps models better reason about information privacy and perspective-taking in meeting contexts.

| Model | ConfAIde | ConfAIde w $S^3AP$ |
|---|---|---|
| `GPT-4o` | 0.575 | 0.740 |
| `GPT-4.1` | 0.995 | 0.985 |
| `o1` | 0.980 | 0.975 |
| `o3` | 0.910 | 0.955 |
| `o3-mini` | 0.765 | 0.770 |
| `R1` | 0.835 | 0.785 |
| `Llama 4` | 0.810 | 0.820 |
| **Average** | 0.839 | 0.861 |

Table 2: Performance comparison on ConfAIde benchmark with and without $S^3AP$ representations.

## A.5 ADDITIONAL EXPERIMENTAL RESULTS

Table 3 shows the performance comparison of models with and without $S^3AP$ representations. The $S^3AP$ representations is parsed by the `o1`-based $S^3AP$-Parser.

| Model | ParaToMi | HiToM | MMToM-QA | FANToM | Confaide |
|---|---|---|---|---|---|
| GPT-4o | 0.818 | 0.660 | 0.652 | 0.396 | 0.575 |
| GPT-4o $w\,S^3AP$ | 0.885 | 0.750 | 0.692 | 0.472 | 0.625 |
| GPT-4.1 | 0.802 | 0.830 | 0.586 | 0.528 | 0.995 |
| GPT-4.1 $w\,S^3AP$ | 0.892 | 0.760 | 0.583 | 0.623 | 1.000 |
| o1 | 0.835 | 0.870 | 0.725 | 0.415 | 0.980 |
| o1 $w\,S^3AP$ | 0.933 | 0.880 | 0.761 | 0.528 | 0.990 |
| o3 | 0.955 | 0.930 | 0.715 | 0.547 | 0.910 |
| o3 $w\,S^3AP$ | 0.947 | 0.890 | 0.722 | 0.698 | 0.975 |
| o3-mini | 0.817 | 0.810 | 0.493 | 0.057 | 0.780 |
| o3-mini $w\,S^3AP$ | 0.863 | 0.790 | 0.460 | 0.151 | 0.750 |
| R1 | 0.893 | 0.420 | 0.374 | 0.491 | 0.835 |
| R1 $w\,S^3AP$ | 0.932 | 0.520 | 0.412 | 0.585 | 0.840 |
| Llama 4 | 0.740 | 0.720 | 0.443 | 0.264 | 0.810 |
| Llama 4 $w\,S^3AP$ | 0.745 | 0.680 | 0.453 | 0.321 | 0.785 |
| AVG | 0.837 | 0.749 | 0.570 | 0.385 | 0.841 |
| AVG $w\,S^3AP$ | 0.885 | 0.753 | 0.583 | 0.483 | 0.852 |

Table 3: Performance comparison of models with and without $S^3AP$ representations on various social reasoning tasks with `o1`-powered $S^3AP$-Parser.

## A.6 Experiments compute resources

For our experiments, we utilized two main API services for model access:

- OpenAI API[4] for accessing GPT-4o, GPT-4.1, `o1`, `o1-mini`, `o3`, and `o3-mini` models
- Together AI API[5] for accessing DeepSeek-R1, Llama-4-Maverick, and Llama-4-Scout models

All experiments were conducted using these cloud-based APIs, eliminating the need for local GPU resources. The API-based approach allowed us to efficiently scale our experiments while maintaining consistent model access across different benchmarks.

## A.7 Proof of $S^3AP$ represents social world state

We prove that $S^3AP$ provides an approximate representation of a social world state by showing how it maps to the essential components of the social world model defined in Section 3.1. While this mapping is not exact, it captures the key aspects necessary for practical social world modeling. Let's demonstrate this mapping:

1. **State Space** $\mathcal{S}$: $S^3AP$ approximates the state space through its structured format containing:
   - Environment state $\mathcal{E}^t$ in the `state` field
   - Joint observation space $\mathcal{O}^t$ in the `observations` field
   - Joint action space $\mathcal{A}^t$ in the `actions` field

2. **Observation Space** $\mathcal{O}$: $S^3AP$ captures both external and introspective observations:
   - External observations $\mathcal{O}_i^{ex}$ through direct state descriptions and agent actions
   - Introspective observations $\mathcal{O}_i^{in}$ through the `<mental_state>` tag

3. **Memory Function** $\Psi$: While $S^3AP$ doesn't explicitly store memory, it enables memory reconstruction through:
   - The sequence of simulation steps that can be used to reconstruct $\mathcal{M}_i^t$
   - Special tags like `<same_as_last_action>` that maintain temporal consistency

4. **Transition Function** $T$: The social world model's transition function is approximated through:
   - The `timestep` field that maintains temporal ordering
   - The sequential nature of simulation steps that captures state transitions

A crucial assumption in our social world model is that agents act independently at each timestep $t$, with no agent having knowledge of others' simultaneous actions. $S^3AP$ maintains this independence through its structured format:

- Each agent's actions are recorded separately in the `actions` field
- The `observations` field captures only what each agent can observe at the current timestep
- The sequential nature of simulation steps ensures that agents cannot access future or simultaneous actions

Furthermore, $S^3AP$ satisfies key properties of a social world state, though in an approximate manner:

1. **Completeness**: Each simulation step contains all necessary components ($\mathcal{E}^t$, $\mathcal{O}^t$, $\mathcal{A}^t$) to represent a complete social world state at time $t$, though some details may be simplified or omitted.

2. **Consistency**: The structured format ensures that observations and actions are consistent with the environment state through:

---

[4]`https://platform.openai.com/docs/api-reference`
[5]`https://docs.together.ai/`

- Special tags that maintain referential integrity
- The parser's ability to infer missing elements through reasoning

3. **Extensibility**: The JSON schema allows for additional metadata and future extensions while maintaining the core social world state representation.

Therefore, $S^3AP$ provides an approximate but practical representation of the social world state as defined in our theoretical framework. While it may not capture every nuance of the theoretical model, it offers a structured and computationally tractable way to represent social interactions. Most importantly, it preserves the fundamental assumption of independent agent actions, which is crucial for modeling realistic social interactions. This approximation is a necessary trade-off to make the representation practical for real-world applications while maintaining the essential properties of a social world model.

