# OpenReview forum: "Social World Models: Universal Structured Representations for Social Reasoning"
_ICLR.cc/2026/Conference — Submitted to ICLR 2026_

### Official Review · Reviewer_pRnY · 2025-10-27

**Soundness:** 2
**Presentation:** 3
**Contribution:** 2
**Rating:** 2
**Confidence:** 4

**Summary:**

This paper introduces Social World Models (SWMs) and a structured representation format called S3AP (Structured Social Simulation Analysis Protocol).

The central idea is to convert free-form social narratives—such as dialogues or stories—into structured tuples describing agents’ states, actions, and mental states, inspired by a Dec-POMDP formulation.
The authors claim that this structured intermediate representation can improve large language models’ (LLMs) social reasoning abilities across multiple theory-of-mind benchmarks (ToMi, ParaToMi, FANToM, etc.) and enhance interactive decision-making on the SOTOPIA platform.

Experimental results show consistent gains over Chain-of-Thought and several previous ToM-specific baselines.

**Strengths:**

The paper addresses an important and underexplored goal: enhancing social reasoning in LLMs through more structured world representations.

The writing and figures are clear, making the proposed representation easy to follow.

The empirical section covers both static and interactive reasoning, which helps connect theory-of-mind evaluation with agentic applications.

**Weaknesses:**

1. **Limited methodological originality**
   While the paper is well presented and motivated, the core mechanism of S3AP mainly reformats social narratives into structured state–observation–action tuples before reasoning.
   This design improves clarity but remains close to a **ReAct-style prompting modification**, rather than introducing a fundamentally new reasoning or learning paradigm.
   The contribution therefore feels more representational than algorithmic, raising questions about whether the improvements stem from true modeling advances or prompt restructuring.

2. **Missing comparisons with modern reasoning frameworks**
   If S3AP is proposed as a general reasoning enhancement, it should be compared against stronger and more recent **agentic reasoning architectures**—such as **LLM-Debate** [1], **AFlow** [2], **DyFlow** [3]—which represent the current frontier of structured, workflow-based reasoning.
   These systems explicitly model dynamic reasoning steps, efficiency, and control flow.
   Without such baselines, it is difficult to assess whether S3AP offers real advantages beyond static prompt organization.

3. **Limited analysis across model scales**
   Most experiments rely on large proprietary models (o1, GPT-4o, R1), which already possess strong social reasoning abilities.
   To demonstrate generality, it would be important to evaluate **different model sizes and families**, e.g., small and medium Qwen models, where social reasoning is notably weaker.
   Such evidence would clarify whether S3AP genuinely enhances reasoning robustness rather than amplifying capabilities of already strong models.

---

**References**
[1] *LLM-Debate: Improving Factuality and Reasoning in Language Models through Multiagent Debate*, ICLR 2024.
[2] *AFlow: Automating Agentic Workflow Generation*, ICLR 2025.
[3] *DyFlow: Dynamic Workflow Framework for Agentic Reasoning*, NeurIPS 2025.

**Questions:**

My main questions align with the issues raised in the Weaknesses section.
I would especially appreciate the authors’ clarification on the intended methodological novelty of S3AP, its differences from existing reasoning frameworks, and how broadly it can generalize across models of different sizes.

While I remain skeptical about the current contribution, I am genuinely interested in the authors’ perspective and open to revising my assessment after reading a thoughtful and detailed rebuttal.

---

> ### Author Response · Authors · 2025-11-24
>
> We thank Reviewer pRnY for the detailed feedback and the opportunity to clarify our methodological novelty and baseline comparisons.
>
> ### 1. On Methodological Novelty (vs. ReAct-style):
>
> > "This design improves clarity but remains close to a ReAct-style prompting modification, rather than introducing a fundamentally new reasoning or learning paradigm."
>
> While it may seem that S3AP and React-style frameworks are similar, they are **fundamentally and conceptually** very different. Our approach makes a theoretical contribution by formalizing a social world-modeling framework that defines (1) latent environment state; (2) per-agent observations, including introspective variables (beliefs, goals, emotions); (3) memory states and memory-update functions; and (4) grounded joint action transitions. These components constitute a learnable state space and transition structure, not a prompting heuristic.
>
> If we conceptualize an S3AP-parser and Social World Model (e.g., via the ForeseeAndAct algorithm that we propose) as external tools, then using an LLM to call the SWM to anticipate the next step could be seen as following the ReAct framework. However, using a broad interpretation, any algorithm, such as RAG or simple two-agent frameworks, can be framed as following the ReAct framework, since ReAct's key contribution is a prompting protocol for chaining reasoning with external actions. Furthermore, ReAct does not address the fundamental issue that LLMs tend to treat context as absolute ground truth and do not naturally "question" their input representations. Without the structural constraints of S3AP, a model simply processes observations as reality, and our formalism forces the explicit decoupling of reality from different agents’ own perception and belief, a necessary step for social reasoning.
>
>
> Our contribution is twofold and distinct:
>
> 1. A Novel Formalism (S3AP): We introduce S3AP as a POMDP-inspired formalism to represent the full multi-agent social world state. This is not about an agent's own actions or observations, but about understanding the entire social environment, including other agents' partial observations and hidden mental states. The representation itself is not an agentic framework, and all of our experiments in Section 4 aim to show that a general-purpose, structured social world representation (S3AP) can be automatically induced from free-form text to significantly improve social reasoning.
>
> 2. A Social Agent Algorithm (FORESEEANDACT): We introduce a planning algorithm (Fig 6) that uses the S3AP formalism. And even in this agentic setting, our focus is on simulating the next social world state for better planning, rather than using action to obtain extra information for better task completion in the ReAct framework.
>
> Overall, while the distinction from ReAct is important, our primary methodological novelty is not about "prompting innovation" but about the **representational advance** itself. We also show this induction is far more efficient than other SOTA methods (Fig 5) and, critically, that this single formalism serves as a robust foundation for building interactive Social World Models (SWMs) for planning.

---

> > ### Author Response · Authors · 2025-11-24
> >
> > ### 2. On Missing Baselines (AFlow, DyFlow, LLM-Debate):
> >
> > > "...it should be compared against stronger and more recent agentic reasoning architectures—such as LLM-Debate [1], AFlow [2], DyFlow [3]..."
> >
> > We thank the reviewer for this point and agree that it is important to differentiate our work from these powerful frameworks. However, their objective is fundamentally different from ours.
> >
> > - **AFlow/DyFlow** are task decomposition frameworks. They model a workflow of sub-tasks to help a single agent solve complex logical problems (e.g., coding, math).
> >
> > - **LLM-Debate** is a collaborative reasoning framework. It models a debate process where multiple agents converge on a single factual answer.
> >
> > - **S3AP / SWM** is a social state formalism. It is designed to represent the hidden mental states of others (Theory of Mind) to enable better social interaction (e.g., Sotopia).
> >
> > The most relevant and correct baselines are other frameworks specifically designed for Theory of Mind (ToM) and modeling other agents. We do compare against these SOTA ToM methods: **AutoToM** and **Thought Tracing (TT)**, as shown in Figure 5 and Section 4.3. Our results show that S3AP achieves superior performance (0.91 avg) over these direct baselines, while also being 10-40x more computationally efficient for reasoning models.
> >
> > Nevertheless, to demonstrate the empirical difference, we evaluated those frameworks against S³AP on a random sample of 100 tasks from ParaToMi (Note that DyFlow requires training, and it works similarly to AFlow, so we only experimented with AFlow here). As shown in the table, S³AP outperforms all agentic baselines while maintaining a fraction of the computational cost. While AFlow and LLM-Debate require significantly more calls to converge on a solution, S³AP achieves better social reasoning accuracy with a fixed, efficient 2-step process.
> >
> > | Base Model | LLM-Debate (9 calls) | AFlow (3 calls) | S³AP (Ours) (2 calls) |
> > | --- | --- | --- | --- |
> > | **GPT-4o** | 85% | 67% | 91% |
> > | **R1** | 91% | 92% | 95% |
> > | **o1** | 92% | 85% | 93% |
> >
> > ### 3. On Limited Model Scales:
> >
> > > "Most experiments rely on large proprietary models (o1, GPT-4o, R1)... To demonstrate generality, it would be important to evaluate different model sizes and families, e.g., small and medium Qwen models, where social reasoning is notably weaker."
> >
> > Respectfully, this is factually incorrect, as our evaluation spans a wide range of models, including top-tier proprietary models (GPT-4o, o1), high-performing open-source models (R1, Llama 4), and smaller models with notably weaker reasoning (03-mini), as shown in Tables 1 and 3.
> >
> > This wide range supports the generalizability of our finding that S3AP genuinely enhances reasoning robustness across different models of different levels of openness. As shown in Table 1, S3AP provides significant gains for these exact models: a **+15.6 point boost** for Llama 4 on FANTOM (0.264 → 0.415) and a **+10.3 point boost** for 03-mini on ToMi (0.863 → 0.960).
> >
> > We agree that verifying performance on the Qwen family specifically, across multiple orders of magnitude, strengthens the claim of generalizability.
> > To address this directly, we conducted additional experiments (paraToMi task; randomly sample 100 test instances) using the Qwen3 suite, ranging from the lightweight 0.6B model to the flagship 80B model. As shown below, our method (S³AP) consistently outperforms standard Chain-of-Thought (CoT) regardless of model scale. We find that small models gain modestly while large models see substantial boosts, revealing a capability threshold needed to fully leverage structured guidance. We also find that S³AP can close scale gaps, enabling a 32B model to surpass the baseline performance of an 80B model.
> > | Model Name | CoT | S³AP |
> > | --- | --- | --- |
> > | **Qwen3-80B** | 77% | 91% |
> > | **Qwen3-32B** | 75% | 88% |
> > | **Qwen3-7B** | 44% | 45% |
> > | **Qwen3-0.6B** | 48% | 50% |
> >
> > We will include the new findings and results in our updated draft. Please let us know if you have any additional questions. If you think we have sufficiently addressed your points, we kindly ask that you consider updating your score.

---

### Official Review · Reviewer_jkJz · 2025-10-30

**Soundness:** 2
**Presentation:** 2
**Contribution:** 2
**Rating:** 6
**Confidence:** 2

**Summary:**

This paper introduces S³AP (Structured Social Simulation Analysis Protocol), a novel structured representation formalism designed to capture the implicit dynamics of social interactions. The authors argue that current AI systems struggle with social reasoning due to the lossy and biased nature of free-text narratives. S³AP addresses this by structuring social world states into components such as environment state, agent observations, actions, and mental states, inspired by a POMDP framework.

**Strengths:**

The authors provide extensive empirical validation across a diverse set of tasks (five static benchmarks and one interactive platform) and multiple state-of-the-art LLMs, demonstrating consistent and often significant performance improvements.
The paper successfully extends S³AP from static analysis to interactive agents via Social World Models, showing tangible benefits in a complex multi-agent environment (SOTOPIA).

**Weaknesses:**

While effective in SOTOPIA with a limited simulation horizon (N=1), the authors acknowledge that tracking mental states grows combinatorially with the number of agents and timesteps. The practical scalability for long, multi-agent interactions remains a significant, unaddressed challenge.
The evaluation is confined to text-based simulations and benchmarks. It is unclear how S³AP would handle real-world noisy data, where social cues are far more subtle and complex.
The S³AP schema is a fixed, pre-defined template. This may limit its ability to adapt to or discover entirely new structures of social interaction that are not captured by the current schema.
For interactive settings with more than two agents or longer horizons, how do you propose to manage the combinatorial explosion of possible mental states?
How would you extend the S³AP framework to incorporate multi-modal inputs (e.g., visual scenes, audio) to create a more grounded social world model?

**Questions:**

See weakness above.

---

> ### Author Response · Authors · 2025-11-24
>
> We thank Reviewer jkJz for their feedback and for acknowledging our extensive empirical validation. We address the primary concerns below.
>
> > While effective in SOTOPIA with a limited simulation horizon (N=1), the authors acknowledge that tracking mental states grows combinatorially with the number of agents and timesteps. The practical scalability for long, multi-agent interactions remains a significant, unaddressed challenge.
>
> We agree that scalability is the key challenge for all multi-agent world models, and we acknowledge this in Appendix A.1. This is a common challenge for most existing works on social reasoning, including AutoToM [1] and Thought Tracing [2]. And Figure 5 demonstrates that S³AP already achieves substantially better accuracy and efficiency compared to recent ToM-specific methods like Thought Tracing [1] and AutoToM [2].
>
> Extending our ForeseeAndAct algorithm to horizons of N>1 introduces a unique challenge due to recursive belief nesting (modeling Agent A's prediction of Agent B's future beliefs), which necessitates complex architectural innovations, such as Social-MCTS or sparse state pruning, that constitute their own distinct research questions. However, S³AP establishes the necessary representational foundation for such future scaling.
>
>
> [1] Zhang et al. (2025) AutoToM: Scaling Model-based Mental Inference
> via Automated Agent Modeling, https://arxiv.org/pdf/2502.15676
>
> [2] Kim et al. (2025) Hypothesis-Driven Theory-of-Mind Reasoning for Large Language Models, https://arxiv.org/abs/2502.11881
>
>
>
>
> > The evaluation is confined to text-based simulations and benchmarks. It is unclear how S3AP would handle real-world noisy data, where social cues are far more subtle and complex. The S3AP schema is a fixed, pre-defined template. This may limit its ability to adapt to or discover entirely new structures of social interaction that are not captured by the current schema.
>
> This is a correct observation of a **deliberate design choice**. The S3AP schema is indeed "fixed" because it is “principled”, drawing its components directly from the well-established POMDP formalism.
>
> The advantage of this principled structure is its *generality*. The same POMDP-inspired components (state, observation, action, mental state) proved effective across *six different benchmarks*, covering diverse tasks from false-belief (ToMi) to multi-party dialogue (FANTOM) and interactive negotiation (SOTOPIA). Furthermore, the structure has great room for extension with designing customized symbolic tags, as we mentioned in Section 3.2.
>
> The social cues in benchmarks such as FANTOM and SOTOPIA are already highly subtle and complex. The fact that powerful state-of-the-art models struggle significantly on these tasks out of the box shows that these benchmarks provide important tests of social reasoning capabilities. We agree that adapting to raw, noisy real-world data is important, and we see future work testing S³AP in live Human-AI interaction settings, where the schema could be extended to handle the "open-world" noise of unstructured social dynamics.
>
>
> > How would you extend the S3AP framework to incorporate multi-modal inputs (e.g., visual scenes, audio) to create a more grounded social world model?
>
> Our current focus is on establishing the S3AP formalism and demonstrating its utility in the text domain, which remains the standard setting for the vast majority of existing social reasoning and ToM benchmarks.
> That said, we view S3AP as a natural foundation for multimodal extensions. The POMDP-inspired structure is inherently extensible: for instance, the observations field in the S3AP schema (Appendix A.2) can be augmented to encode non-textual inputs (e.g., "observations": { "agent_A": "sees <person_B_frowning>" }"), which can then be produced by appropriate multimodal foundation models.
>
> Please let us know if you have any additional questions. If you think we have sufficiently addressed your points, we kindly ask that you consider updating your score.

---

> > ### Comment · Reviewer_jkJz · 2025-11-27
> > **Response from reviwers**
> >
> > Thanks for your response. I will maintain my score.

---

> > > ### Author Response · Authors · 2025-11-27
> > >
> > > We thank the reviewer for their continued engagement. We would like to politely inquire if the new experiments and findings we recently posted (specifically regarding model scaling and parser fidelity) help address your remaining concerns about scalability and robustness.
> > >
> > > You can view the detailed results in the latest response to the Reviewer EWPY (https://openreview.net/forum?id=vC6DGcAdWR&noteId=4ETde6bpuk)
> > >
> > > We are happy to answer any further questions.

---

### Official Review · Reviewer_EWPY · 2025-10-30

**Soundness:** 3
**Presentation:** 3
**Contribution:** 3
**Rating:** 6
**Confidence:** 4

**Summary:**

The paper proposes a POMDP-inspired schema that encodes, per timestep, environment state, agent actions, and agent mental states parsed from raw narratives. An LLM parser maps text to SAP tuples; feeding these tuples to LLMs improves performance on social reasoning benchmarks. The authors also induce a “social world model” to predict future actions/mental states, yielding higher task success in multi-turn SOTOPIA. These gains hold across model families and settings, with larger benefits in some cooperative scenarios. Overall, it appears that explicit structure bridges narrative understanding and interactive planning.

**Strengths:**

The paper provides clear, and general formalism for social dynamics.

SOTA improvements on diverse belief tracking tasks.

The study provides extension from offline reasoning to interactive planning with a learned/prompted world model.

There are model agnostic benefits seen across LLM scales and thorough eval suite.

**Weaknesses:**

While the paper and proposed system introduces a complex solution to an interesting problem, this proposed system seems to introduce many moving parts. For instance, a parser to convert raw text to the schedule, the integration of this schema to LLM prompts, and an algo to simulate a social world model. This complexity is interesting but further raises questions about reliability. In particular, how accurate is the S3AP in parsing itself? There is no reporting on evolution of the parser’s fidelity, so in this case, it would be hard to assess if errors in the structure's representation might propagate. This approach might benefit from an analysis of parsing quality or examples of typical parse outputting. Additionally, it is unclear if using this ‘universal’ claim should hold as it is untested beyond text and two agent simulations. It is also unclear if it scales to multimodal, many agents, or long horizon settings.

One other concern is practicality, reliance on large-model calls could be brittle or expensive in practice. While the authors acknowledge that fully simulating multistep mental state dynamics is computationally intensive, needing simplified one step lookahead in practice. This constraint would suggest that the method, as is, might struggle with very long or even real time interactions due to this complexity.

It appears that the success of the parser and improvements in reasoning largely leverage pre-trained LLMs abilities. The system reforms the input problem for the LLM by structuring it, instead of fundamentally learning social reasoning from scratch. This might mean the approach ‘inherits’ all the biases and knowledge gaps of the underlying models. So, for example, the parser might fill in some said inferred mental states or context that are plausible but not actually in the narrative, depending on the LLMs prior knowledge. Thus, the pipeline could occasionally introduce hallucinated facts or inconsistent agent states if the LLM overinterprets the text. Just to cover this base, the paper could benefit from examples or error analysis where it fails; for instance, does it ever insert incorrect assumptions about an agent’s belief that lead to wrong answers? One recommendation is a critical analysis of failure cases to establish the method’s reliability.

From a cognitive science perspective, it is interesting that the model is able to explicitly represent others’ beliefs and goals. However, though not critical, the work does not compare these representations to human social reasoning. Incorporating classic TOM tasks, even as case studies, would provide valuable context and even highlight whenever the model avoids/reproduces common human reasoning pitfalls.

**Questions:**

How would the approach handle inputs beyond text based narratives? For ex, in a real multimodal interaction, do you envision extending this framework to include perceptual elements or do you think the narrative description approach still be used as an intermediate? Also, does it scale to scenarios with more agents or longer running interactions?
Would the authors please clarify how the social model is trained or used within the agent? Is it a learned model or a procedural roll out using the LLM at each step? The paper does mention inducing the world model from the S3AP but it would be useful if we could know if this involved any additional learning?
The authors found that having a world model seems to yield different advantages in cooperative vs competitive settings.Could the authors please elaborate on the differences observed? So, did the world model improve success rates much more in cooperative than in competitive ones and how?
Did the authors compare S3AP’s structure representation to simpler prompting strategies like knowledge graphs etc. for these tasks? The authors showed one comparison with a CoT prompt baseline on some benchmarks where S3AP did better, but would be useful to know if any intermediate representation provides similar gains.

---

> ### Author Response · Authors · 2025-11-24
>
> We thank Reviewer EWPY for the positive and constructive review. We address the weaknesses and questions below
>
> ### 1. On Misunderstanding the S3AP-Parser's Role (Weakness 1):
>
> We believe there is a misunderstanding regarding the "moving parts".
>
> > **Reviewer:** While the paper and proposed system introduces a complex solution to an interesting problem, this proposed system seems to introduce many moving parts. For instance, a parser to convert raw text to the schedule, the integration of this schema to LLM prompts, and an algo to simulate a social world model. This complexity is interesting but further raises questions about reliability.
>
> The S3AP-Parser is only used for static, third-person narrative benchmarks (e.g., ToMi, FANTOM, as shown in §4) to convert free-form text into our S3AP representation. And here we aim to show how structured social world representations can benefit social reasoning.
>
> However, in the experiments involved in the interactive SOTOPIA benchmark (§5), we **do not use a parser**. In SOTOPIA, the agent's actions and observations are already available in a structured format. The SWM's role here is not to parse, but to *generate the next social world state* (p(S^{t+1}|S^t, A^t)) based on the current S3AP state and the agent's action. This pipeline is simpler and demonstrates the SWM's *planning capabilities*, not its parsing ability.
>
> ### 2. On Parser Accuracy and Error Analysis (Weakness 1 & 3):
>
> > In particular, how accurate is the S3AP in parsing itself? There is no reporting on evolution of the parser’s fidelity, so in this case, it would be hard to assess if errors in the structure's representation might propagate. This approach might benefit from an analysis of parsing quality or examples of typical parse outputting.
>
> We agree that parser fidelity is important, as shown in section 4.2.
>
> - Parser Accuracy: While a full-scale human annotation of all parsed states is prohibitively difficult, we manually annotate 64 instances from the ParaToMi task parsed by o1. We find a low error rate of only 9% in the S3AP-Parser's output.
>
> - Error Analysis: We also conducted a detailed error analysis on the downstream task. We found that 79.7% of downstream reasoning errors on ParaToMi stem from "social context parsing failures." We believe this finding *strengthens our claim*, as it precisely highlights the importance of the "representation" component. It shows that when parsing fails, reasoning often fails, validating that a correct representation (which S3AP provides) is the key.
>
> > Additionally, it is unclear if using this ‘universal’ claim should hold as it is untested beyond text and two agent simulations. It is also unclear if it scales to multimodal, many agents, or long horizon settings.
>
> > How would the approach handle inputs beyond text based narratives? For ex, in a real multimodal interaction, do you envision extending this framework to include perceptual elements or do you think the narrative description approach still be used as an intermediate? Also, does it scale to scenarios with more agents or longer running interactions?
>
> We appreciate this point. We agree that our interactive experiments (Sotopia) are limited to dyadic (two-agent) interactions. This is primarily due to the lack of reliable, high-quality benchmarks for multi-party (3+) interactive social simulations.
> While interactive benchmarks are dyadic, our static benchmarks (Table 1, e.g., ParaToMi) frequently require modeling complex mental states **among 3 or more participants**. S³AP successfully handles these scenarios, demonstrating that the formalism itself is not structurally limited to pairs.
>
> For multimodality, while our current operationalization is text-based to align with standard benchmarks, the S³AP formalism is designed to be general and input-agnostic. We see future work adapting S³AP to multimodal settings by mapping perceptual inputs, such as VLM-generated scene descriptions or audio affect tags, directly into the observations field of our schema (e.g., observations: { agent_A: "sees <visual_cue_frowning> (or referring to specific image)" }). This would allow the same structure to function on grounded, multimodal data without changing the core framework.

---

> > ### Author Response · Authors · 2025-11-24
> >
> > > This might mean the approach ‘inherits’ all the biases and knowledge gaps of the underlying models... Just to cover this base, the paper could benefit from examples or error analysis where it fails; for instance, does it ever insert incorrect assumptions about an agent’s belief that lead to wrong answers? One recommendation is a critical analysis of failure cases to establish the method’s reliability.
> >
> > We agree with the reviewer that this is a valid concern. To address this, we included an error analysis discussing such cases (lines 314–322 and 343–354), including whether incorrect assumptions about agents’ beliefs can lead to wrong answers.
> > Since manually parsing text into S3AP representations across all benchmarks would be infeasible due to scale, we therefore focus on evaluating how the generated representations from different models affect downstream reasoning performance, as a proxy for parser accuracy.
> > To this end, we conducted extensive experiments that swap the parser and reasoner models. As shown in Figure 4, a weaker model (03-mini) can produce S3AP representations that significantly improve a stronger model’s (o1) reasoning accuracy, from 84% to 94%. This demonstrates that the benefits come from the structured representations themselves, despite some underlying errors in the representations.
> >
> > > From a cognitive science perspective, it is interesting that the model is able to explicitly represent others’ beliefs and goals. However, though not critical, the work does not compare these representations to human social reasoning. Incorporating classic TOM tasks, even as case studies, would provide valuable context and even highlight whenever the model avoids/reproduces common human reasoning pitfalls.
> >
> > We agree that such comparisons to classic ToM tasks and human performance on them would be interesting. In fact, ToMi and ParaToMi are inspired by classic ToM tasks (Sally-Anne test), and our approach is designed to capture the same underlying structure of belief and goals in humans. Our work focuses primarily on evaluating how structured S3AP representations affect LLM reasoning performance across large-scale, diverse benchmarks. Incorporating full human-model comparisons would require a separate, carefully controlled study, and we view this as an exciting direction for future work.
> >
> > > Would the authors please clarify how the social model is trained or used within the agent? Is it a learned model or a procedural roll out using the LLM at each step? The paper does mention inducing the world model from the S3AP but it would be useful if we could know if this involved any additional learning?
> >
> > The SWM is not a fine-tuned/learned model. We "induce" the world model by using a zero-shot LLM to perform the procedural simulation step (i.e., predicting the next S3AP state given the current state and an action).
> >
> >
> > > The authors found that having a world model seems to yield different advantages in cooperative vs competitive settings.Could the authors please elaborate on the differences observed? So, did the world model improve success rates much more in cooperative than in competitive ones and how?
> >
> > We observed that while the Social World Model (SWM) improves performance in both settings, the mechanism of this improvement differs, as in competitive tasks, SWM enables strategic exploration of hidden mental states, whereas in cooperative tasks, it drives information efficiency.
> >
> > - In Competitive (Bargaining): The SWM allows the agent to *"Predict the interlocutor's goal"*. The SWM simulates that the seller ("Zane") internally *"wants the sale."* This lookahead insight allows our agent to adopt a *"More aggressive strategy"* (offering $175 instead of $180) and succeed. SWM is more important in this case since actively predicting the seller’s mental states is crucial for reaching a better deal.
> >
> > - In Cooperative (Mutual Friend): The SWM helps the agent *"Reinforce their own goal"* by simulating both agents' mental states. This refines a general query ("know the host?") into a *"More targeted question"* ("have you spent any time in Minnesota?").

---

> > > ### Author Response · Authors · 2025-11-24
> > >
> > > >  Did the authors compare S3AP’s structure representation to simpler prompting strategies like knowledge graphs etc. for these tasks? The authors showed one comparison with a CoT prompt baseline on some benchmarks where S3AP did better, but would be useful to know if any intermediate representation provides similar gains.
> > >
> > > This is an excellent question that touches on two points: (1) representations like KGs, and (2) other intermediate representations.
> > >
> > > - Knowledge Graphs: Conceptually, the S³AP formalism can be viewed as a specialized, dynamic Knowledge Graph, where nodes represent agents and edges represent epistemic relations (e.g., Agent A → Believes → P) rather than static semantic facts. However, we distinguish S³AP from generic Knowledge Graph prompting because standard KGs typically model static entity relationships (e.g., “Alice is a Teacher”). Furthermore, S³AP functions as a POMDP-inspired state-transition system, explicitly tracking how new observations update latent variables (beliefs, goals) over time. Thus, S³AP is optimized specifically for the partial observability and dynamic nature of social interaction that generic KG usually don’t capture.
> > >
> > > - Other Intermediate Representations (SOTA ToM Methods): We respectfully note that we **do compare S3AP to state-of-the-art explicit Theory of Mind (ToM) reasoning methods**, which also rely on advanced intermediate representations. As shown in Figure 5 and Section 4.3, we compare S3AP against **SOTA ToM methods AutoToM and Thought Tracing (TT)**.
> > >
> > >   - Our results demonstrate that S3AP achieves the highest average performance (0.91) on the ParaToMi benchmark compared to all baselines, including TT (0.83) and AutoToM (0.88).
> > >
> > >   - Crucially, S3AP is also *7-20x more computationally efficient*, requiring only 1-2 LLM calls, whereas TT requires \~15 calls and AutoToM requires \~40 calls.
> > >
> > >   - As the reviewer noted, "large-model calls could be... expensive," and this cost becomes *prohibitively expensive*, preventing us from running these complex baselines on all benchmarks.
> > >
> > > Therefore, our method is not only stronger than the CoT baseline but also substantially outperforms the most relevant and powerful SOTA intermediate representations for ToM in both accuracy and efficiency. We will clarify this in the updated version.
> > >
> > > Please let us know if you have any additional questions. If you think we have sufficiently addressed your points, we kindly ask that you consider updating your score.

---

> > > > ### Comment · Reviewer_EWPY · 2025-11-26
> > > >
> > > > Thank you for the polite and comprehensive rebuttal in scope. One thing to note is that the rebuttal is primarily relying on existing evidence from the paper rather than new data/experiments. As a result the overall initial (largely positive) assessment of the paper is unchanged. The authors have reinforced why their approach is interesting and how they thought through some of the challenges but the key concerns still remain. Specific points are outlined below:
> > > >
> > > > Re: complexity/reliability of the s3ap system. The core reliability concern is minimized in the author’s response. Describing S3Ap as a “necessary trade-off” and explaining its design, does not demonstrate robustness in practice. Though the significant performance improvements are good and encouraging, this does not necessarily guarantee reliability in all cases or domains.
> > > >
> > > > Re: parser Fidelity and Error Prop: Thank you for the note about the parser fidelity issue. However, the solution offered is not entirely satisfying. It would be great to demonstrate any systemic fix beyond choosing a high qual LLM parser. There is no reported parser accuracy metric or additional experiment in the rebuttal that shows the reduced error prop. The weakness still remains and the rebuttal does not fully align/fix with the original concern.
> > > >
> > > > Re: generalizability to multimodal or multi agent settings: The explanation for multi agent gen is convincing. However the multimodal generalization remains more speculative. The authors do not provide concrete experiments or examples w/images/videos when they are asserting extensibility. Thus, the original concern and question about whether this system would work beyond text narratives is not actually fully resolved.
> > > >
> > > > Re: dependency of Large model prior/hallucinations: The response only partially addresses the original concern. Yes, it is understandable that the gpt4 level models were needed to get the reported performance and the transparency here is appreciated. However, this, still, does not resolve whether or not the method would still be effective w/smaller or open source models. For hallucination risk, the authors do not offer new safeguards against hallucination beyond the structured output prompt or quantitative evidence or similar for this. Thus, the original concern that the system might occasionally introduce incorrect information due to LLM priors is still valid.

---

> ### Author Response · Authors · 2025-11-27
>
> We thank Reviewer EWPY for the engagement. We appreciate the push for concrete data. We have run new experiments for the Qwen-series models (also in the responses for Reviewer pRnY), and we have newly added an evaluation for parser fidelity/error propagation.
>
> **Re: Dependency on Large Models & Effectiveness with Open Source Models**
>
> We agree that verifying performance across multiple models is important. New Experiment (also in response to reviewer pRnY): We conducted additional experiments on the Qwen 3 suite (ParaToMi task; 100 randomly sampled test instances), ranging from the lightweight 0.6B model to the flagship 80B model.
>
> Results: As shown below, S^3AP consistently outperforms standard Chain-of-Thought (CoT). Most notably, S^3AP allows a smaller model to punch above its weight class; the 32 B model with S^3AP (88%) outperforms the 80B model using CoT (77%). It's also interesting to see that small models gain modestly while large models see substantial boosts, indicating a capability threshold needed to fully leverage structured guidance.
>
> | Model Name | CoT | S³AP |
> | --- | --- | --- |
> | **Qwen3-80B** | 77% | 91% |
> | **Qwen3-32B** | 75% | 88% |
> | **Qwen3-7B** | 44% | 45% |
> | **Qwen3-0.6B** | 48% | 50% |
>
> **Re: Parser Fidelity and Error Propagation**
>
> We agree that assessing error propagation is important. However, we respectfully note that evaluating the quality of the generated S³AP representation is fundamentally challenging. Much like evaluating Chain-of-Thought (CoT) quality, there is no single unique "correct" intermediate representation. Furthermore, different models often segment the narrative into different time steps, generating S³AP updates at different granularities, which makes direct automated comparison difficult.
>
> Nevertheless, to provide the requested quantitative evidence, we **newly** ran experiments on the ParaToMi set (100 randomly selected instances). To establish a baseline, we manually annotated/corrected the S³AP representations generated by O3 to create a "Ground Truth" set. We then used an LLM-as-a-judge (GPT-5) to score the S³AP generated by other models against this ground truth.
>
> **Judge Prompt Used:**
> > **Task:** Evaluate whether the generated socialized context correctly captures what each state actually aligns with in the story.
> >
> > **Task Story:** {story}
> > **Ground Truth Socialized Context:** {gt_formatted}
> > **Generated Socialized Context to Evaluate:** {gen_formatted}
>
> **Results:**
>
> | Model | Overall Parser Score (0-1) | ParaToMi Improvement (averaged from Fig 4) |
> | :--- | :--- | :--- |
> | **O3-mini** | **0.707** | **+7.64%** |
> | **O1** | 0.650 | +4.44% |
> | **GPT-4o** | 0.620 | +6.74% |
> | **Llama-4** | 0.591 | +5.36% |
> | **DeepSeek-R1** | 0.490 | +3.74% |
>
> We observe that the Overall Parser Score aligns closely with the ParaToMi Improvement (except O1), suggesting that higher-quality representations indeed lead to better reasoning outcomes.
>
> However, a surprising and significant finding is that *smaller models (like O3-mini) can achieve high parsing scores*, even surpassing larger models like GPT-4o or O1. This suggests that the capability to parse social dynamics is distinct from general question-answering ability. This is promising as smaller, efficient models can generate high-quality structures that drive significant performance gains for stronger/bigger models.
>
>
> We hope our new experiments and clarifications sufficiently address the reviewer's concerns regarding open-source models' applicability and parser fidelity.

---

### Official Review · Reviewer_LdM5 · 2025-11-01

**Soundness:** 2
**Presentation:** 3
**Contribution:** 2
**Rating:** 2
**Confidence:** 4

**Summary:**

This paper presents a structured textual representation, S3AP,  for the social world. The observation space is extended with the agent’s internal mental states, like beliefs, goals, moral values, and emotions. The action space is extended to recall memory, reflect on past actions, and update beliefs. This paper performs multiple experiments across existing benchmarks, showing the effectiveness of the proposed S3AP.

**Strengths:**

- The representation can be constructed using the same prompt across different LLMs.
- The proposed representation is compatible across multiple LLMs.
- S3AP achieves high performance with low LLM calls.

**Weaknesses:**

- While eq 1 and 2 illustrate the computation of the social world model (SWM), it implies that the method relies on the internal world modeling capability of LLMs rather than building any world model, so the results can be highly affected by the LLM’s capability.
- The baselines only include chain-of-thought, which is not the most powerful ToM reasoning method on these benchmarks. It is still hard to conclude how the proposed method performs against other explicit ToM reasoning methods like BIP-ALM [1] or Thought Tracing.
   - [1] Jin et al. MMToM-QA: Multimodal theory of mind question answering. ACL 2024.
- As the authors mention, the current input representations AI systems learn from “mention only salient events, omit explicit mentions of mental states…”. However, the proposed method heavily depends on LLMs’s capability to parse the mental states, which is trained from static text too. This limitation is unavoidable for the proposed method, which raises concerns about the validity of both S3AP and the SWM.

**Questions:**

- What’s the accuracy of the parsed mental states? How can the accuracy of the mental states in the posed results affect the performance?
- Although Figure 8 illustrates the final utterance, it is unclear about the SWM's internal mechanisms. Could you provide some specific examples to explicitly demonstrate how the model generates At(−i)​ and predicts the future state given the current state and action, and how that affects performance?
- When prompting the LLM to parse free-form narratives into S3AP, what is the procedure for text truncation as shown in Figure 3? Specifically, how is it determined which sentences or text segments accurately belong to time step t?

---

> ### Author Response · Authors · 2025-11-24
>
> We thank Reviewer LdM5 for their feedback. We address the primary concerns below.
>
> ### 1. On Missing Baselines (BIP-ALM, Thought Tracing):
>
> > The baselines only include chain-of-thought, which is not the most powerful ToM reasoning method on these benchmarks. It is still hard to conclude how the proposed method performs against other explicit ToM reasoning methods like BIP-ALM [1] or Thought Tracing.
>
> We respectfully note that we **do compare S3AP to state-of-the-art explicit Theory of Mind (ToM) reasoning methods**, including Thought Tracing (TT; [1]) and AutoToM [2] (which is essentially a more “advanced” version of BIP-ALM)
>
> Cocretely, here is a summary of the specific methods compared and how S³AP improves upon them: (1) Thought Tracing (TT): A dedicated ToM reasoning framework. S³AP outperforms TT in accuracy (0.91 vs 0.83 on ParaToMi) while being \~7x more efficient (2 calls vs \~15 calls). (2) AutoToM (Advanced BIP-ALM): We compared against AutoToM, which is a more advanced iteration of the BIP-ALM architecture cited. S³AP achieves higher accuracy (0.91 vs 0.88) while being \~20x more efficient (2 calls vs \~40 calls). (3) General Reasoning Baselines: We also compared against Vanilla, Few-Shot, and Chain-of-Thought (CoT) across multiple model families.
> Note that the high computational cost of methods like AutoToM makes them prohibitively expensive for scalable deployment with reasoning models.
>
> [1] Kim et al. (2025) Hypothesis-Driven Theory-of-Mind Reasoning for Large Language Models, https://arxiv.org/abs/2502.11881
> [2] Zhang et al. (2025) AutoToM: Scaling Model-based Mental Inference
> via Automated Agent Modeling, https://arxiv.org/pdf/2502.15676
>
> ### 2. On LLM Dependence and Parser Accuracy:
>  > While eq 1 and 2 illustrate the computation of the social world model (SWM), it implies that the method relies on the internal world modeling capability of LLMs rather than building any world model, so the results can be highly affected by the LLM’s capability.
>
> > As the authors mention, the current input representations AI systems learn from “mention only salient events, omit explicit mentions of mental states…”. However, the proposed method heavily depends on LLMs’s capability to parse the mental states, which is trained from static text too. This limitation is unavoidable for the proposed method, which raises concerns about the validity of both S3AP and the SWM.
>
> Equations 1 and 2 are theoretical definitions of a Social World Model (SWM). They formally define what components are necessary to model social dynamics (latent environment states, introspective variables, etc.), regardless of the actual model used to compute them. **These equations are not inherently tied to LLMs**; they could, in theory, be parameterized by future neuro-symbolic systems or other learned representations.
>
> In this paper, we operationalize our theory by using S³AP to bridge the gap between latent knowledge and active reasoning. Previous research demonstrates that while models possess social knowledge (performing well on benchmarks like Social IQa [1]), they struggle to apply it in complex generative settings [2, 3]. S³AP addresses this by forcing the explicit characterization of these implicit variables, effectively scaffolding the model to surface "long-tail" mental states it otherwise overlooks. Thus, our formalism does not require a perfect internal world model, but rather provides the necessary structure to activate the social capabilities LLMs already possess but fail to utilize.
>
> We completely agree with the reviewer that empirical results are inevitably affected by the backbone LLM’s capability. To explicitly test this dependency, we investigated cross-model performance (Figure 4) and found that representations generated by a notably weaker model (o3-mini) still boosted a stronger model’s (o1) accuracy from 84% to 94%. This finding shows that while backbone capability matters, the S³AP formalism itself provides a structural advantage that is robust to these variations. The fact that a weaker parser can guide a stronger reasoner validates that our formulation contributes distinctly from the raw capabilities of the underlying model.
>
> [1] Sap et al. (2019), “SocialIQA: Commonsense Reasoning about Social Interactions” (https://arxiv.org/abs/1904.09728)
>
> [2] West et al. (2023), "The Generative AI Paradox"  (https://arxiv.org/abs/2311.00059)
>
> [3] Yerukola et al. (2024), "Generative Evaluation of Non-Literal Intent Resolution in LLMs" (https://arxiv.org/abs/2405.08760)

---

> ### Author Response · Authors · 2025-11-24
>
> > “What’s the accuracy of the parsed mental states? How can the accuracy of the mental states in the posed results affect the performance?”
>
> The exact accuracy of the parsed mental states would rely on human annotation. With the 64 instances we annotated for the O1-produced S3AP on the ParaToMi tasks (Section 4.2), the error rate is only 9%. We would like to point out that parser accuracy is important. We conducted a detailed error analysis and found that 79.7% of reasoning errors on ParaToMi come from "social context parsing failures". This actually highlights the importance of the “representation” part of social reasoning.
>
> Creating a robust social parser is non-trivial. Unlike standard Information Extraction (which retrieves explicit facts), social parsing requires inferring latent variables, such as beliefs, unstated goals, and emotional nuances, that are not always explicitly written in the text. By formalizing this challenge, our work identifies "Social Parsing" as a distinct, high-impact research direction, opening the door for future work toward training dedicated "Social Parsers" specifically optimized to map unstructured interactions into structured social world models.
>
> >Although Figure 8 illustrates the final utterance, it is unclear about the SWM's internal mechanisms. Could you provide some specific examples to explicitly demonstrate how the model generates At(−i)​ and predicts the future state given the current state and action, and how that affects performance?
>
> The internal mechanism is our FORESEEANDACT algorithm (Figure 6). It is an inference-time planning algorithm. As shown in Figure 8:
>
> 1. The Agent LLM proposes a candidate action (e.g., "I would like to buy this table for $20").
>
> 2. The SWM LLM: takes the current S3AP state and the agent's potential action (“No, the table sells for $200”) to simulate the next (future) social world state. This includes predicting the interlocutor's resulting mental activities (“The seller won't budge on price, so I'll just leave.”) as well as the agents’ observation (“The buyer looks unhappy”)
>
> 3. The Agent LLM receives this simulated future state ("The buyer looks unhappy") and refines its action based on this lookahead.
>
> > When prompting the LLM to parse free-form narratives into S3AP, what is the procedure for text truncation as shown in Figure 3? Specifically, how is it determined which sentences or text segments accurately belong to time step t?
>
> For simplicity, we let the LLM decide which piece of text should belong to a certain state in S3AP.
>
> Please let us know if you have any additional questions. If you think we have sufficiently addressed your points, we kindly ask that you consider updating your score.

---

### Meta-Review · Area_Chair_Xzu2 · 2026-01-09

**Summary:**

This paper proposes S³AP, a POMDP-inspired structured representation for “social world states” (environment, observations, actions, and mental states) that is automatically parsed from free-form narratives, and then used both to improve static social reasoning benchmarks and to drive a simple “Foresee and Act” social world model on SOTOPIA. Empirically, the authors show consistent gains over a CoT baseline and over ToM-specific methods such as AutoToM and Thought Tracing on ParaToMi, and modest improvements on other ToM datasets and SOTOPIA, with some additional experiments on Qwen models and a parser-fidelity analysis added in the rebuttal. The work is clearly written, and the experimental section is broad and carefully executed.

However, after reading the paper, reviews, and rebuttal, I do not think the contribution meets the ICLR bar. The core novelty is representational: S³AP largely reformats text into a hand-designed schema, and the “social world model” is a zero-shot LLM used procedurally at inference time rather than a learned model with new training principles. This makes the work feel closer to an elaborate prompting / interface design than a fundamentally new modeling framework, echoing concerns from Reviewers pRnY and LdM5 about limited methodological originality and heavy reliance on underlying LLM capabilities. Important conceptual issues also remain insufficiently resolved: the universality claim is not really validated beyond text and dyadic SOTOPIA interactions (EWPY, jkJz), scalability to many agents and long horizons is acknowledged but left open, and despite additional analysis the parser fidelity and error-propagation story still does not convincingly demonstrate robustness or guard against hallucinated mental states. Overall, I see this as an interesting and well-executed system paper, but the idea is not particularly deep or inspiring from a modeling perspective, and the remaining doubts about generality and reliability lead me to recommend rejection.

**Reviewer Concerns:**

Reviewer LdM5.

The rebuttal directly addressed several concrete points: the authors clarified that they *do* compare against strong ToM baselines (Thought Tracing, AutoToM) and provided relative accuracy/LLM-call improvements; they explained that the SWM is implemented as an inference-time “ForeseeAndAct” procedure rather than a separately trained model; and they gave some numbers on parser accuracy (9% error on 64 annotated ParaToMi instances) and an error analysis linking most failures to parsing mistakes.  However, LdM5’s more fundamental concerns remain: the approach is still heavily dependent on backbone LLM world-modelling and mental-state parsing, and the paper does not fully de-risk the validity of S³AP/SWM beyond these capabilities.

Reviewer EWPY.

The authors responded thoroughly to EWPY’s concerns about system complexity, parser fidelity, generalization, and large-model dependence: they added parser-accuracy/error-propagation experiments and new Qwen3-family results showing S³AP gains across model scales, and clarified how S³AP can be extended in principle to multimodal inputs.  Nevertheless, EWPY explicitly notes that core issues are not resolved: reliability is demonstrated mainly by performance gains rather than robustness guarantees; parser fidelity still lacks a systemic fix; multimodal and multi-agent generalization remains speculative; and the risk that the system inherits LLM priors and hallucinations is only partially mitigated.

Reviewer pRnY.

The rebuttal offers a detailed conceptual argument that S³AP is more than “ReAct-style prompting,” positions it as a POMDP-inspired state formalism plus a planning algorithm, and adds targeted comparisons against AFlow and LLM-Debate on ParaToMi, where S³AP is reported to be more accurate and cheaper. It also corrects the claim about limited model scales and supplements this with new Qwen3 experiments.  However, pRnY’s central skepticism about methodological originality is, in my view, only partially alleviated: the work still reads primarily as a representational / interface innovation that restructures inputs for existing LLMs, rather than a clearly new learning paradigm.

Reviewer jkJz.

The authors acknowledge jkJz’s concerns about scalability (many agents, long horizons) and confinement to text, and argue that these are general open problems; they present S³AP’s fixed schema as a principled POMDP-derived design that has worked across several text benchmarks and sketch how it could be extended to multimodal observations.  But there are no new experiments beyond two-agent, short-horizon, text-only settings, so the practical questions about scaling to richer, noisier social environments remain essentially outstanding.

**Reviewer Scores:**

Reviewer LdM5. This reviewer was clearly negative in the initial review, emphasizing limited methodological originality and heavy dependence on backbone LLM capabilities. The rebuttal clarified some details but did not change the reviewer's core view. I expect LdM5 to keep the original (negative) score unchanged.

Reviewer pRnY. pRnY’s main concern is that the method is essentially an interface/prompting scheme rather than a genuinely new modeling framework. The rebuttal tries to position S³AP as a POMDP-style state formalism, but in substance, the approach still looks the same. I think pRnY would also keep the original (negative) score.

Reviewer EWPY. This reviewer is more positive overall, and appreciates the breadth of experiments and the motivation, but explicitly notes remaining concerns about reliability, parser error propagation, and generalization beyond dyadic text interactions even after the rebuttal. Given that they acknowledge improvements but still flag these issues, I expect the reviewer would maintain the original (mildly positive but cautious) score rather than increase it.

Reviewer jkJz. jkJz is the other relatively positive reviewer, but the comments also highlight unresolved questions about scalability (more agents, longer horizons) and the breadth of the “universal” claim. The rebuttal responds conceptually but does not provide new experimental evidence on these points. I would expect jkJz to keep their initial score as well, rather than shifting upward.

---

### Decision · Program_Chairs · 2026-01-26

Reject